# Changes in the Dimensions of Lignocellulose Nanofibrils with Different Lignin Contents by Enzymatic Hydrolysis

**DOI:** 10.3390/polym12102201

**Published:** 2020-09-25

**Authors:** Jae-Hyuk Jang, Noriko Hayashi, Song-Yi Han, Chan-Woo Park, Fauzi Febrianto, Seung-Hwan Lee, Nam-Hun Kim

**Affiliations:** 1Department of Forest Biomaterials and Engineering, College of Forest and Environmental Sciences, Kangwon National University, Chuncheon 24341, Korea; jhtojh@naver.com (J.-H.J.); songyi618@kangwon.ac.kr (S.-Y.H.); chanwoo8973@kangwon.ac.kr (C.-W.P.); lshyhk@kangwon.ac.kr (S.-H.L.); 2Forestry and Forest Products Research Institute, Ibaraki 300-1244, Japan; hayashin@ffpri.affrc.go.jp; 3Institute of Forest Science, Kangwon National University, Chuncheon 24341, Korea; 4Department of Forest Products, Faculty of Forestry and Environment, IPB University (Bogor Agricultural University), Bogor 16680, Indonesia; febrianto76@yahoo.com

**Keywords:** lignocellulose nanofibril, enzymatic hydrolysis, endoglucanase, pretreatment

## Abstract

Changes in the dimensions of lignocellulose nanofibrils (LCNFs) with different lignin contents from betung bamboo (*Dendrocalamus asper*) by enzymatic hydrolysis using endoglucanase (EG) were investigated. Lignin contents were adjusted from 3% to 27% by NaClO_2_/acetic acid treatment, and LCNFs were prepared using a wet disk-mill (WDM). The dimensions of the LCNFs significantly decreased with decreasing lignin content and increasing EG addition. With increasing EG content, the average diameter of the LCNFs significantly decreased, even though they contained parts of hemicellulose and lignin. The crystal structure showed the typical cellulose I structure in all samples, but the intensity of the diffraction peak slightly changed depending on the lignin and EG contents. The crystallinity index (CrI) values of the LCNFs increased a maximum of 23.8% (LCNF-L27) under increasing EG addition, regardless of the lignin content. With the EG addition of three times the LCNF amount, LCNF-L3 showed the highest CrI value (59.1%). By controlling the composition and structure of LCNFs, it is expected that the wide range of properties of these materials can extend the property range available for existing materials.

## 1. Introduction

Recently, lignocellulose nanofibril (LCNF) has been attracting attention in various research areas due to its surface characteristics and impressive mechanical properties [1,2,3,4,5,6,7,8,9]. LCNF contains lignin and hemicellulose and can be obtained from lignocellulose through a mechanical defibrillation process [4,5,6,7,8,9]. LCNF suspension has a relatively lower viscosity than holocellulose nanofibril (HCNF) and pure cellulose nanofibril (PCNF) because of the presence of hydrophobic lignin; therefore, it might have higher dispersibility and excellent affinity with hydrophilic and hydrophobic polymers [6]. It is highly suitable for the utilization of LCNF to various functional composites [10]. In order to prepare LCNF with high efficiency, a pretreatment process is essential because lignocellulose a has tight hierarchical structure and biomass recalcitrance [5,11,12]. Enzymatic hydrolysis is considered an effective method for adjusting the aspect ratio and reducing the dimensions of LCNFs [13,14,15,16].

Enzymes work well under mild process conditions. The use of enzymes in the hydrolysis of cellulose is therefore more effective than the use of inorganic catalysts. Cellulose biodegradation has generally been considered to involve only three types of hydrolytic enzymes: endoglucanase (EG), cellobiohydrolase (CBH), and β-glucosidase (BGL) [17,18,19,20,21]. EGs randomly cleave the internal β-1, 4-glucosidic links, CBHs act on the free ends of cellulose polymer chains, and BGLs hydrolyze cellobiose and other water-soluble cellodextrins to glucose [22]. It is expected that the enzyme pretreatment can loosen cell wall structure due to partial cellulose degradation, thereby improving the efficiency of mechanical defibrillation into LCNF. Pääkkö et al. (2007) [23] obtained PCNF with well-controlled diameters in the nanometer range and high aspect ratios by combining enzymatic hydrolysis and mechanical shearing by high-pressure homogenization. They performed an enzymatic treatment with endoglucanase before passing the pulp slurry through a microfluidizer. Such enzymatic hydrolysis is less aggressive than acid hydrolysis, and it allows for the selective hydrolysis of the noncrystalline cellulose, which facilitates mechanical disintegration [13,24,25].

However, up to now, most studies on the enzymatic hydrolysis of CNFs have focused on bleached pulp or pure cellulose materials without lignin for starting materials. In this study, we investigated the effect of enzymatic hydrolysis when using EG on the changes in the dimensions of LCNFs from betung bamboo with different lignin contents.

## 2. Materials and Methods:

### 2.1. Materials

Betung bamboo was obtained from Arboretum in Bogor Agricultural University, Indonesia, and cutter milled to a 0.2 mm size. Sodium chlorite (NaClO_2_) and acetic acid for delignification, along with other chemicals, were guaranteed reagent grade from commercial suppliers and used without further purification. EG (NS44019) with an enzyme activity of 4500 ECU/g was purchased from Novozymes, Denmark. Sodium dihydrogen phosphate and disodium hydrogen phosphate heptahydrate were purchased from Wako Pure Chemical Industries, Ltd. (Tokyo, Japan) and Nakarai Chemicals, Ltd. (Tokyo, Japan), respectively.

### 2.2. Delignification

Holocellulose from betung bamboo was prepared using an NaClO_2_/acetic acid treatment described in the literature [26,27]. Cutter-milled bamboo powder was suspended to be 1%, and delignification was conducted at 70 °C for 1 h with the successive addition of 0.3 g of NaClO_2_ and 0.1 mL of acetic acid per gram of oven-dried bamboo. A series of LCNFs with different lignin contents was prepared by changing the reaction cycle (1, 2, 3, and 4 times) and the amounts of reagents used. The resulting products were repeatedly washed with distilled water until the pH became neutral.

### 2.3. Preparation of LCNF

The delignified wood powders for LCNF were suspended at 0.5 wt% concentration. The suspensions were subjected to mechanical defibrillation using a wet disk-mill (WDM) (Supermasscolloider MKCA6-2, Masuko Sangyo Co. Ltd., Kawaguchi, Japan). The clearance between the upper and lower disks was set to be 200 μm from the zero position, at which point the disks began to rub, and the rotational speed was 1800 rpm. The operation was repeated for 8 passes. The total milling time was recorded, and then the WDM time (min/kg) of each pass was calculated based on the solid weight of LCNFs. The diameter of individual fibers was measured at least 300 times on each sample by ImageJ software (National Institute of Health, Bethesda, MD, USA).

### 2.4. Enzymatic Hydrolysis by EG

For enzymatic hydrolysis, the LCNF suspensions were diluted to 0.1 wt% suspensions (50 mL) with a phosphate buffer. The phosphate buffer was prepared according to the literature [28]. Stock solutions of monobasic and dibasic sodium phosphate (0.2 M) were obtained from sodium dihydrogen phosphate and disodium hydrogen phosphate heptahydrate, respectively. The two types of solutions were thoroughly mixed by magnetic stirring for 30 min at room temperature. The EG was added into the LCNF suspensions with ratios of 1/1, 1/2, and 1/3 (LCNF/EG). Then, the mixture was sonicated for 2 h and stirred overnight at 30 °C. This process was repeated two times. The mixtures were centrifuged at a speed of 10,000 rpm, and the resulting supernatants were separated.

### 2.5. Characteristics

Characterization was performed using the supernatants after the centrifugation of the LCNFs before and after enzymatic hydrolysis. Polarizing optical microscopy (POM, Nikon Optiphot-Pol, Nikon Corp., Tokyo, Japan) and TEM (JEM-2000EX, JEOL, Japan) were used to observe the morphology of the hydrolyzed products. To observe TEM images, a drop of a diluted bamboo LCNF suspension was deposited on a carbon-coated grid and allowed to dry at room temperature. The length and diameter of individual LCNFs were measured at least 50 times from POM images and TEM images, respectively.

The crystalline characteristics of LCNFs were determined using XRD (RINT 2000, Rigaku Corporation, Tokyo, Japan). For X-ray analyses, disks that were 1 cm in diameter, 0.8 mm in thickness, and 0.1 g in weight were prepared from the freeze-dried samples. Ni-filtered Cu Kα radiation (λ = 0.1542 nm) was employed at an accelerating voltage of 200 kV and a current of 40 mA. The diffracted intensity (I) was determined in the 2θ range of 5°–40° at a rate of 2°/min. The crystallinity index (CrI) was calculated using Segal’s method (formula (1)) [29].
(1)$$\mathrm{Crystallinity}\;\mathrm{index}\left(\%\right)=\;\frac{\left(I_{200}-I_{\mathrm{amor}}\right)}{I_{200}}\times 100$$
where I_200_ is the crystalline intensity and I_amor_ is the amorphous intensity.

## 3. Results and Discussion

### 3.1. Lignin Contents of LCNFs

The lignin content was adjusted by NaClO_2_/acetic acid delignification. The lignin content decreased from 27% to 3% with increasing NaClO_2_/acetic acid repetition. Depending on the lignin content of the LCNFs, the sample names were denoted as LCNF-L27, -L23, -L18, -L9, and -L3, as shown in Table 1. Kumar et al. (2013) [27] reported that NaClO_2_/acetic acid delignification for 8 h applied to a range of cellulosic biomass types (switchgrass, poplar, corn stover, and pine sawdust) removed more than 90% of the lignin. They adjusted the lignin content of hinoki cypress by using NaClO_2_/acetic acid delignification with different reaction times. They reported that with increasing reaction time (0, 10 min, 4 h, 7 h, and 8 h), the lignin content decreased from 27.2% to 6.5%, whereas the composition of the constituent sugars of hemicellulose was not significantly changed.

### 3.2. Morphology and Size of Insufficiently Defibrillated Products

Products that were insufficiently defibrillated because of their less enzymatic hydrolysis or mechanical defibrillation, which are considered to be micrometer scale, could be observed in the low-magnification setting of the optical microscope. Figure 1 shows polarizing optical micrographs of insufficiently defibrillated products obtained from LCNFs with different lignin contents after enzymatic treatment using different EG contents. With decreasing lignin content, the size of fibers was significantly decreased. The dimension and occurrence frequencies of the insufficiently defibrillated products were significantly decreased with increasing EG contents. In particular, the fibers of micron-scale were hardly observed in EG-treated LCNF-L9 and LCNF-L3 at ratios of 1/2 and 1/3 (LCNF/EG), respectively. The effect of lignin and EG contents on the length distribution of the insufficiently defibrillated products is described in Figure 2. In the all samples, the length of the insufficiently defibrillated products was reduced, and their distributions of length became narrower with decreasing lignin content and increasing EG content. This phenomenon was more distinct in the LCNFs with lower lignin contents. The average lengths measured from polarizing optical micrographs are summarized in Table 2. The fiber length was drastically reduced by EG additions. In the case of LCNF-L27, the fiber length decreased from 118.6 to 71.4 mm by addition of EG. Henriksson et al. (2007) [25] prepared CNF from bleached wood sulphite pulp via an enzymatic treatment using EG, acid hydrolysis, and high-pressure homogenization. They reported that the fiber length was reduced by addition of EG, and the extent of fine material increased. It was considered that the addition of an enzyme promoted the fiber delamination and thus improved the finer morphology.

### 3.3. Morphology and Diameter Distribution of LCNFs

Figure 3 shows TEM images of LCNF without EG treatment and LCNF treated by EG with different contents of lignin and EG. Without EG treatment, LCNF-L27 showed the biggest diameter among the LCNFs. It was observed that the dimensions of LCNF were significantly decreased with decreasing lignin content and increasing EG contents. The effects of lignin and EG contents on the diameter distribution of LCNFs are shown in Figure 4. In all samples, the diameters of the LCNFs decreased and the diameter distributions narrowed significantly with decreasing lignin and increasing EG contents, respectively. In the samples treated with EG at ratios of 1/2 and 1/3 (LCNF/EG), LCNFs showed a diameter of less than 50 nm regardless of lignin content.

The average diameters of the LCNFs are summarized in Table 3. The diameters of the LCNF were decreased with increasing EG contents, even though they contained some lignin. In particular, in LCNF-L27, the diameter of the LCNF drastically decreased from 200.2 to 17.3 nm by EG treatment. Furthermore, in the LCNF-L9 and LCNF-L3, most of the diameters in all the samples were estimated to be less than 13 nm, which is considered to be the diameter of a single cellulose microfibril. Espinosa et al. (2017) [30] performed an enzymatic pretreatment and then high-pressure homogenization to obtain LCNF from unbleached wheat straw soda pulp (lignin content of 17.7%). It was reported that the average diameter of the LCNFs after the enzymatic process was about 14.5nm. Therefore, it can be considered that size-controllable CNFs can be produced by adjusting the lignin content and EG addition in the EG-assisted mechanical grinding process.

### 3.4. Crystalline Characteristics of LCNFs

Figure 5 shows X-ray diffractograms of LCNF without EG treatment and LCNF treated by EG at different contents of lignin and EG. All the diffractograms show peaks at about 2θ = 18.5° (I_amor_) and 22.5° (I_200_), which are considered to represent the typical cellulose I structure, thus indicating that the crystal integrity was maintained [31]. However, the I_200_ diffraction patterns were significantly changed by lignin content and EG addition. That is, the I_200_ reflection peaks of the LCNFs sharpened with decreasing lignin contents and increasing EG contents. It can be considered that the degradation of the amorphous zones of the LCNFs was proceeded by EG, resulting in an increased crystallinity [32]. The CrI values of the EG-treated bamboo LCNFs that were calculated based on Segal’s method [29] are given in Table 4. Untreated LCNFs exhibited much lower CrI values (32.2%). However, the CrI values of the LCNFs drastically increased with decreasing lignin content and EG concentration. Qing et al. (2013) [15] reported that CNFs obtained from bleached eucalyptus kraft pulp after WDM treatment followed by enzymatic hydrolysis showed a CrI of 60% compared with a pulp fiber crystallinity of 55%. The increase of crystallinity confirmed the hypothesis that the selected enzyme partially digested amorphous regions. Liu et al. (2020) [16] reported that the bleached bagasse kraft pulp was subjected to EG pretreatment with a wide range of enzyme dosages (1, 10, 30, 60, and 120 IU/g) and subsequent mechanical grinding to prepare tunable CNFs with a wide range of sizes. The CrI was increased from 46.1% to 60.2% with an increase in the enzyme dosage from 0 to 120 IU/g in the pretreatment.

## 4. Conclusions

In this study, the effect of EG treatment on the changes in the dimensions of LCNFs with different lignin contents was investigated. The dimensions of the LCNFs significantly decreased with decreasing lignin content and increasing EG content. When the amount of EG added was more than two times the amount of LCNF during EG treatment, LCNFs showed a diameter of less than 50 nm regardless of lignin content. In the LCNFs with less than 9% lignin content, most of the diameters in all the samples were estimated to be less than 13 nm, which is considered to be the diameter of a single cellulose microfibril. The crystal structure showed the typical cellulose I structure in all samples, but the intensity of the diffraction peak slightly changed depending on the lignin content and EG addition. The CrI of the LCNF-L3 treated with the EG at a ratio of 1/3 (LCNF/EG) was the highest value (59.1%). By controlling the composition and structure of LCNFs, it is expected that the wide range of properties of these materials can extend the property range available for existing materials.

## Figures and Tables

**Figure 1 polymers-12-02201-f001:**
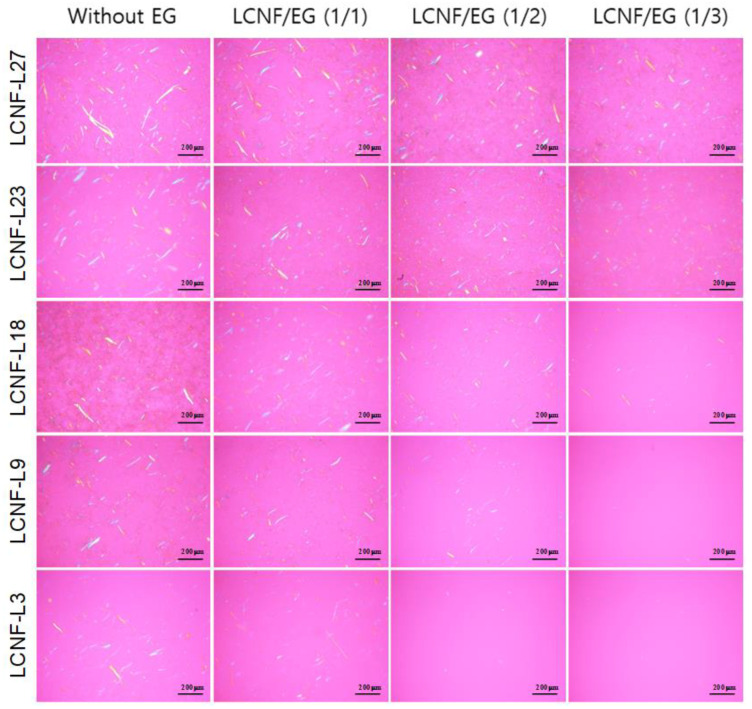
Polarizing optical micrographs of the insufficiently defibrillated products obtained from lignocellulose nanofibrils (LCNFs) with different lignin contents after enzymatic treatment using different endoglucanase (EG) contents.

**Figure 2 polymers-12-02201-f002:**
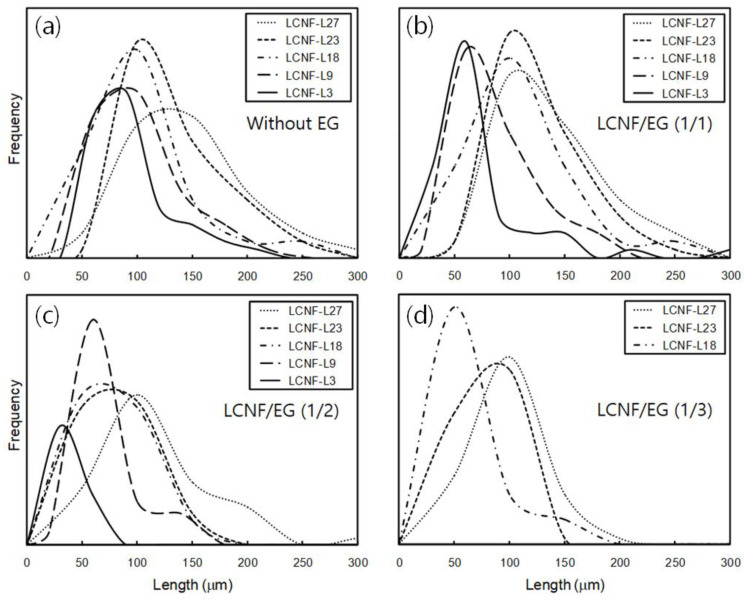
Diameter distribution of the insufficiently defibrillated products with different lignin contents after enzymatic treatment using different EG contents; (**a**) without EG, (**b**) LCNF/EG (1/1), (**c**) LCNF/EG (1/2), (**d**) LCNF/EG (1/3).

**Figure 3 polymers-12-02201-f003:**
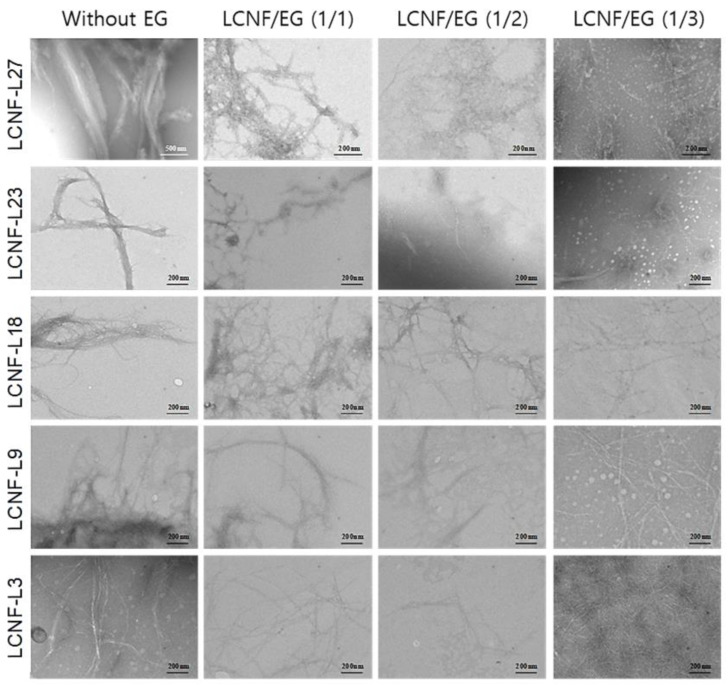
TEM images of LCNFs with different lignin contents after enzymatic treatment using different EG contents.

**Figure 4 polymers-12-02201-f004:**
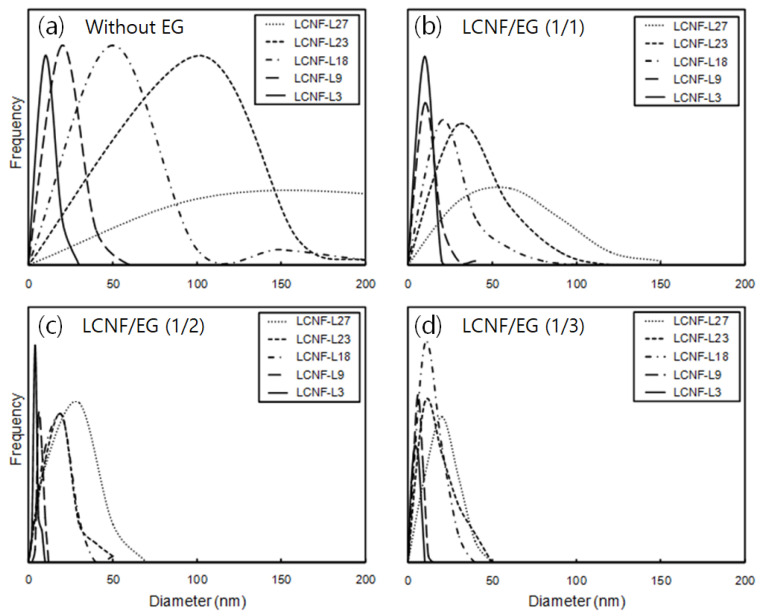
Diameter distribution of LCNFs with different lignin contents after enzymatic treatment using different EG contents; (**a**) without EG, (**b**) LCNF/EG (1/1), (**c**) LCNF/EG (1/2), (**d**) LCNF/EG (1/3).

**Figure 5 polymers-12-02201-f005:**
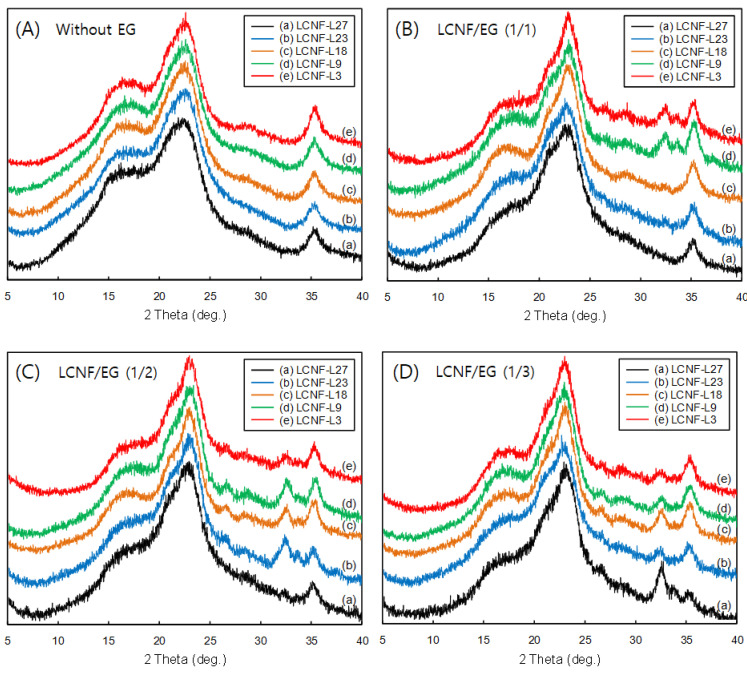
X-ray diffractograms of LCNFs with different lignin contents after enzymatic treatment using different EG contents; (**A**) without EG, (**B**) LCNF/EG (1/1), (**C**) LCNF/EG (1/2), (**D**) LCNF/EG (1/3).

**Table 1 polymers-12-02201-t001:** Sample codes of lignocellulose nanofibrils (LCNFs) with different lignin content degrees.

Sample Code	LCNF-L27	LCNF-L23	LCNF-L18	LCNF-L9	LCNF-L3
Lignin contents (%)	27	23	18	9	3

**Table 2 polymers-12-02201-t002:** Average lengths and standard deviations of the insufficiently defibrillated products with different lignin contents after enzymatic treatment using different EG contents (units: μm).

Sample Code	Without EG	Ratio of Substrate/Enzyme (LCNF/EG)		
		1/1	1/2	1/3
LCNF-L27	118.6 ± 41.8	104.4 ± 32.3	89.5 ± 32.9	71.4 ± 20.1
LCNF-L23	101.4 ± 33.9	95.2 ± 29.9	57.3 ± 16.2	56.1 ± 15.9
LCNF-L18	77.8 ± 26.5	80.1 ± 33.5	53.9 ± 22.1	40.9 ± 10.6
LCNF-L9	74.8 ± 24.1	62.6 ± 26.0	44.2 ± 17.0	N/A
LCNF-L3	59.6 ± 15.6	48.0 ± 20.4	21.4 ± 8.4	N/A

**Table 3 polymers-12-02201-t003:** Average diameter and standard deviations of LCNFs with different lignin contents after enzymatic treatment using different EG contents (units: nm).

Sample Code	Without EG	Ratio of Substrate/Enzyme (LCNF/EG)		
		1/1	1/2	1/3
LCNF-L27	200.2 ± 119.2	49.5 ± 21.5	18.0 ± 8.2	17.3 ± 7.6
LCNF-L23	45.3 ± 31.6	24.9 ± 11.2	16.3 ± 10.0	12.6 ± 6.4
LCNF-L18	20.8 ± 15.8	16.6 ± 9.5	12.1 ± 4.9	9.9 ± 4.0
LCNF-L9	12.3 ± 5.8	7.9 ± 1.8	6.5 ± 1.1	4.4 ± 1.6
LCNF-L3	6.7 ± 2.4	4.4 ± 1.3	3.9 ± 0.7	3.8 ± 1.5

**Table 4 polymers-12-02201-t004:** Crystallinity index (CrI) of the LCNFs with different lignin contents after enzymatic treatment using different EG contents (units: %).

Sample Code	Without EG	Ratio of Substrate/Enzyme (LCNF/EG)		
		1/1	1/2	1/3
LCNF-L27	32.2	38.9	50.0	56.0
LCNF-L23	39.3	44.7	52.7	50.9
LCNF-L18	40.9	47.4	58.3	58.0
LCNF-L9	41.7	48.6	52.7	54.5
LCNF-L3	42.2	54.0	57.7	59.1

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
