# Peer review of "Changes in the Dimensions of Lignocellulose Nanofibrils with Different Lignin Contents by Enzymatic Hydrolysis"

_polymers, 2020, doi:10.3390/polym12102201_

Round 1

Reviewer 1 Report

This manuscript deals with the changes in dimensions of lignocellulose nanofibrils with different lignin contents due to enzymatic hydrolysis. Generally, this is not a new approach, however, the characterisation of the enzymatic impacts of different lignin contents of lignocellulose is quite interesting. The manuscript needs a small revision before publishing.

The conclusion section (e.g. 1-2 sentences) in the abstract is missing.

Materials and Methods

Which methods and devices were used for the determination of diameter and its distribution of LCNF? How many nanofibrils were measured?

Results and Discussion

Line 116-118: As a result, with increasing reaction time (0, 10 min, 4 h, 7 h, and 8 h), the lignin content decreased from 27.2 to 6.5 %, whereas the … significantly changed.

Where can the readers find/verify this information (what is the connection to the LCNF-27, LCNF-23, ...)? Which was the lignin content (e.g. reference sample) without acid delignification process?

Figure 2 shows the diameter distributions of LCNF with different processes and lignin contents of the lignocelluloses. However, in Figure 1 some networks of the nanofibrils can be observed.

How did you prove that your measurements are fine (cf. comments for the Material and Method section)?

Table 2; I guess that you included also the standard deviations of your measurements, please mention this information also in the table heading of Table 2;

Author Response

Response to Reviewer 1 Comments

This manuscript deals with the changes in dimensions of lignocellulose nanofibrils with different lignin contents due to enzymatic hydrolysis. Generally, this is not a new approach, however, the characterisation of the enzymatic impacts of different lignin contents of lignocellulose is quite interesting. The manuscript needs a small revision before publishing.

Point 1. The conclusion section (e.g. 1-2 sentences) in the abstract is missing.

Response 1. First, we’d appreciate your critical review. We add the ‘conclusion section’ in the Abstract. Thank you very much.

Point 2. Which methods and devices were used for the determination of diameter and its distribution of LCNF? How many nanofibrils were measured?

Response 2. Thank you for your comment. We measured the diameter of individual fibers at least 300 times using a software ‘Image J’. We added this information in ‘2.3. Preparation of LCNF’ line 81. Thank you.

Point 3. Line 116-118: As a result, with increasing reaction time (0, 10 min, 4 h, 7 h, and 8 h), the lignin content decreased from 27.2 to 6.5 %, whereas the … significantly changed. Where can the readers find/verify this information (what is the connection to the LCNF-27, LCNF-23, ...)? Which was the lignin content (e.g. reference sample) without acid delignification process?

Response 3. We apologize for the confusion. We cited data from Kumar et al. to refer to the delignification effect by NaClO2/acetic acid treatment. The sentence “As a result, with increasing reaction time (0, 10 min, 4 h, 7 h, and 8 h), the lignin content decreased from 27.2 to 6.5 %, whereas the … significantly changed.” was described like the experimental results of this study. To avoid this confusion, we modified this sentence; “They reported that with increasing reaction time (0, 10 min, 4 h, 7 h, and 8 h), the lignin content decreased from 27.2 to 6.5 %, whereas the … significantly changed.”. Thank you for your critical review.

Point 4. Figure 2 shows the diameter distributions of LCNF with different processes and lignin contents of the lignocelluloses. However, in Figure 1 some networks of the nanofibrils can be observed. How did you prove that your measurements are fine (cf. comments for the Material and Method section)?

Response 4. Thank you for your valuable comment. As you know, it is not easy to measure the diameter of LCNFs perfectly from TEM images because some fibers are tangled like a net. However, we tried to improve the accuracy by measuring as many fibers as possible. Thank you very much. 

Point 5. Table 2; I guess that you included also the standard deviations of your measurements, please mention this information also in the table heading of Table 2;

Response 5. Thank you for your review. We mentioned the standard deviations in the table heading of Table 2. Thank you.

Reviewer 2 Report

This is an interesting study and broadens the experimental control of lignin rich nanocellulosics

Author Response

Response to Reviewer 2 Comments

This is an interesting study and broadens the experimental control of lignin rich nanocellulosics

Response : We’d appreciate your reviewing for this manuscript. Thank you for giving us the opportunity to submit this manuscript to polymers.

Reviewer 3 Report

I consider the article important and interesting from the practical point of view of the manufacturing of lignocellulosic composites, which is a current research issue. I have no comments regarding the text of the article, it is short but clearly written, only in section 2.3 Preparation of LCNF the authors should give WDM times, which as they write were counted. Also, the quality of TEM photos (Fig. 1) should be definitely better. The discussion of the results is correct and is also based on the literature. The conclusions are brief, but show the main results of the research. The bibliography is not very extensive, but quite up-to-date (2 articles from 2020 and 1 from 2019). In conclusion, the article can be published in Polymers Journal as long as the TEM images are improved. It should attract the interest of readers mainly from the point of view of the practical use of the research described in it and the results obtained.

Author Response

Response to Reviewer 3 Comments

I consider the article important and interesting from the practical point of view of the manufacturing of lignocellulosic composites, which is a current research issue. I have no comments regarding the text of the article, it is short but clearly written, only in section 2.3 Preparation of LCNF the authors should give WDM times, which as they write were counted. Also, the quality of TEM photos (Fig. 1) should be definitely better. The discussion of the results is correct and is also based on the literature. The conclusions are brief, but show the main results of the research. The bibliography is not very extensive, but quite up-to-date (2 articles from 2020 and 1 from 2019). In conclusion, the article can be published in Polymers Journal as long as the TEM images are improved. It should attract the interest of readers mainly from the point of view of the practical use of the research described in it and the results obtained.

Point 1. Preparation of LCNF the authors should give WDM times, which as they write were counted.

Response1. First, we’d appreciate your review. Thank you for your valuable comments. We’d modified the ‘Preparation of LCNF’ to explain ‘WDM time’ clearly. Thank you very much.

Point 2. The quality of TEM photos (Fig. 1) should be definitely better.

Response 2. We replaced the Fig. 1 with a higher quality pictures. It will be help readers observe the LCNF morphologies more clearly. Thank you.